# An Arab Jew Reads the Quran: On Isaac Yahuda's Hebrew Commentary on the Islamic Scripture

Mostafa Hussein

LSA Judaic Studies, University of Michigan, Ann Arbor, MI 48104-1608, USA; mostafah@umich.edu

**Abstract:** How did an Arab Jew read the Quran against the backdrop of contradictory ideologies and the rise of key movements, including nationalism, colonialism, and Zionism, in Mandate Palestine? Approaching Isaac Yahuda as an Arab Jew challenges the binary opposition between Arabs and Jews in Zionist discourse, a linkage perceived as inconceivable, and on the other hand, that linkage is asserted, contested, and tested in the context of nationalism. This article also challenges the advancement of Jewish singularity and superiority by exploring how Jewish writers interacted with the Islamic scripture in Mandatory Palestine rather than dismissing it. This article examines Hebrew interpretation of various passages from the Quran that produced an understanding of the Quran that advanced Zionist ideals, including the nationalization of contested religious sites and the consolidation of the indigeneity of Jews in the East. Isaac Yahuda's Hebrew commentary on the Quran challenged his Arab Jewishness in such a divisive nationalist atmosphere in Mandate Palestine. His hybrid background and dynamic connections with both Jews and Arabs enabled him to navigate these turbulent times by invoking the Quran, demonstrating respect for it, and at the same time challenging the understanding of his contemporary Muslims while utilizing German Jewish scholarship on the origins of Islam.

**Keywords:** Islam; Judaism; Quran; Hebrew Bible; Levantinism

## 1. Introduction

The central question of this paper is how an Arab Jew read the Quran in Mandatory Palestine, a politically turbulent period in the history of nationalism and colonialism in the Middle East, begs several questions that would not make such an act of reading an easy thing to do. An Arab Jew reading of the Quran and his commentary on it reflected his theological beliefs as a devout Jew who sought in both the Bible and the Quran ways of enriching the understanding of both texts through comparative linguistic and religious studies. Nevertheless, this intellectual activity equally raised questions regarding the extent to which his scholarly or quasi-scholarly works had a bearing on and relationship to the political struggle for self-determination, predominant intellectual currents within the Hebrew cultural movement as well as within the Naḥḍa (Arab cultural renaissance), and finally, common Orientalist tropes and modes that affected the relations between East and West and penetrated into the East to construct internal Orientalization and stigmatization. It is understandable that the use of the term "Arab Jew" is a controversial one. The application of the term "Arab Jew" to the protagonist of this story, Isaac Yahuda, however, aims at several objectives. The first is to challenge the normative term in Zionist discourse by underscoring the hybridity and the inclusion of Jews within Arab societies (Shenhav 2006; Shohat 2006). The second is to articulate the manner in which a number of Jews were very well versed in Arabo-Islamic culture and pushed back against Zionism's normative assumptions that "Arab" and "Jew" were inherently contradictory and irreconcilable. The third is to explore how Arab Jews viewed their affiliation with Zionism and their struggle within a dominant European movement and their attempts to create a version of Zionism enshrined in the culture of the East and opposed mainstream European Zionists who

wanted to live in the East without being part of it (Shohat 2006, p. 331). The fourth is to underscore "the horizontal relations" between Jews and Arabs based on everyday life, customs, and beliefs; a social and cultural proximity that created an imagined community of belonging (Klein 2014, p. 135). That identity existed in late Ottoman Palestine, prior to the emergence of Zionism and Arab nationalism (Tamārī 2004a, p. 11). However, the escalation in violent confrontations between Jewish and Arab communities during the 1930s against the watershed of a national struggle for self-determination weakened that identity and led to its demise by 1945 (Klein 2014, p. 135). In this article, I posit that Isaac Yahuda's Hebrew interpretation of the Quran as well as its appropriation contributed to the promotion of Zionist ideas, such as the nationalization of religious sites that were under dispute between Jews and Muslims and the consolidation of the indigeneity of Jews in the East. His commentary on the Quran also posed a challenge to his identity as an Arab Jew in the midst of the nationalist tensions in Mandatory Palestine. Yahuda's mixed background and his connections with both Jewish and Arab communities, nevertheless, allowed him to navigate this politically and socially challenging period by using the Quran to contest the understanding of contemporary Muslims while demonstrating respect for the scripture. Additionally, Yahuda's commentary was influenced by the work of German Jewish scholars who had studied the origins of Islam and considered thematic similarities between both texts as biblical borrowings.

## 2. The Centrality of the Sacred Text in the East

With the emergence of modernity, both the Bible and the Quran occupied a central place in the Jewish and Arab modern configurations: for Jews, in the Haskalah (the Jewish Enlightenment) and the Zionist movement; and for Arabs, in the Nahḍa (the cultural awakening movement); and also for Arab Jewish writers who participated in the Nahḍa movement. Both scriptures were seen as cross-cultural linguistic boundaries and served as a prism for the blended culture of the East. Arab intellectuals, whether Jews, Christians, or Muslims, related to the Quran in their preoccupations with selfhood and homeland as well as with the crafting of culture. Buṭrus al-Bustānī, a Christian Arab writer, for instance, used quotations from the Quran in the revitalization of Arab culture, demonstrating "the pluralistic nature of his patriotic platform", which made him vulnerable to criticism by his Christian coreligionists who accused him of inserting "non-Christian references" into his writings (Sheehi 2004, p. 49). Asther Moyal, Murad Farag, and Ya'cob Sanua' used Quranic quotations in their works in their treatment of issues pertaining to relations between Islam and modernity and religious minorities (Behar and Benite 2013, pp. 48–61; Levy 2012, p. 129). For Taha Hussein, known as the doyen of Arabic literature, the Quran appears to be as important to the development of the Egyptian mind as the Bible to the European intellect. He remarks that the European interest in the Bible, a scripture originated in the East, had not hindered Europe's progress; similarly, then, the Quran should not be viewed as an obstacle to progress in the East. According to Hussein's worldview, the pre-eminence of the Quran derives from supplementing the Bible, not contradicting it (Hussein 1958, pp. 248–49). This view is consolidated in Hussein's suggestion to those specializing in Arabic and Islamic studies to devote intellectual energy to acquire linguistic skills in either Hebrew or Aramaic. Also, the Quran proved a useful and instructive text in the dialectics of East and West. In addition to modern Jewish interest in the Quran that was expressed in Arabic writings in the East, there were works in Hebrew on the scriptural text.

Isaac Yahuda's (1863–1941) interaction with the Quran in the late Ottoman era provides a window into what historian Ussama Makdisi has termed "the age of coexistence". His fascination with Arabic from an early age drove him to independent study devoting daily hours to learning the Quran and classical and medieval Arabic literature under the influence of his Palestinian Arabic teachers Ibrahim al-Salfiti and later Muhammad Jarallah who had learned at al-Azhar at the end of the nineteenth century. Muhammad Jarallah taught Arabic at the Alliance Israélite Universelle's school (Ben-Ze'ev 1941, p. 73). Through the prism of Makdisi's ecumenicalism that suggests "the rich diversity of *al-Mashriq* [East] that has

stubbornly refused repeated attempts to reduce the region to one religious hue", Yahuda's writings on Arabo-Islamic culture can be seen to overcome religious differences between Jews and other religious communities and to deplore the religious extremism that enabled the dissemination of negative portrayals of Jews (Makdisi 2019, p. 7). Yahuda's interaction with the Islamic scripture is evident in his editing and commenting on the Arabic work *Sharḥal-Maḍnūn bi-hi ʿala ghayr ahlih* [That which is to be Withheld from Those Unworthy of It], where Quranic verses are brought in to elaborate on the meaning and the function of certain phrases in the cited poetic verses, or in Hebrew (See cited Quranic verses in Al-ʾUbaydī 1913–1915, pp. 22, 24, 36, 38).

*Arab Jewish Intellectuals and Nahḍa*

Recent scholarship has underscored the engagement of a small group of acculturated Arab Jewish writers in the Nahḍa discourse "as a vehicle of Jewish integration into an emerging interconfessional 'modern' class and also to debate the general shared concerns of Arab modernity, helping to shape this discourse in the process". These individuals operated in the multicultural milieu in the Middle East, including Cairo, Beirut, Jerusalem, and Jaffa, and their writings featured rhetorical modes and tropes typical of Nahḍa. Scholars have contextualized the contributions of certain Arab Jewish intellectuals to the Nahḍa discourse and their involvement in the tropes and the themes of modernity and the return to origins within the Arab enlighteners (Levy 2013, p. 304). Among those Arab Jewish writers who were part of the interconfessional modern class in the East and participated in the Nahḍa discourse was Isaac Yahuda.

Along with learning biblical and extrabiblical sources at yeshivot in his youth, Isaac Yahuda showed an interest in Arabic and learned the language at the hands of Muslim and Christian Arab teachers in Jerusalem, his hometown, and further pursued his Arabic studies at the Alliance Israélite Universelle (Ben-Zeʾev 1941, p. 73). Years of learning Arabic language and literature qualified him to become an Arabic teacher at the Alliance and other Jewish schools in Jerusalem in addition to offering Arabic lessons to Ashkenazi Jewish immigrants. His efforts in the dissemination of Arabic were not limited to only teaching but extended to offer services to residents of Jerusalem to write requests in Arabic to the local government (Ben-Zeʾev 1941, p. 73). His move to Cairo in 1906 to make a living through trading in Arabo-Islamic manuscripts and books enabled him to deepen his Arabo-Islamic knowledge and to contribute to the dissemination of that knowledge among Western Orientalists and Arab and Muslim scholars (Ben-Zeʾev 1941). This business allowed him to trade in books and manuscripts in oriental languages, including Hebrew, Arabic, Persian, and Turkish, and brought him closer into the circle of prominent Arab and Muslim intellectuals and scholars involved in the Nahḍa movement such as Ahmad Taymur, Ahmad Zaki, and Ahmad Lutfi al-Sayyid. His expertise in deciphering and reading medieval manuscripts attracted to him a number of distinguished Muslim scholars like Muhammad al-Biblawi, the supervisor of Arabic Literary Revival at the Egyptian National Library (*Dar al-Kutub al-Misriyyah*) and other scholars who took part in the movement of Arabic literary revival in Egypt (Ben-Zeʾev 1941, p. 74).

Isaac Yahuda entertained his Arab Jewishness in seeking connections between Arabic and Hebrew as well as ties between Arabo-Islamic and Jewish cultures, work that garnered the respect and admiration of contemporary Arab Muslim scholars. In a novel treating the social and cultural life of Jerusalem's Jews before the outbreak of World War Two, Yahuda is presented as Yeshurun Deuel, who emerges as a Jewish scholar well versed in Arabo-Islamic texts. His reputation made him an authority in the field to the degree that Muslim scholars at the most highly professional school of Arabic and Islamic studies at the time, Al-Azhar in Cairo, turned to him to resolve the enigmatic content of texts and thus reveal their history and legacy. In the novel, we read:

> "A few days earlier, Deuel received an invitation from al-Azhar University in Cairo asking his appearance to analyze an old manuscript that came to their hands. . . When he arrived at the university, Mr. Deuel received a small book. . .

Deuel read easily the blurry script whose content related to the period where writers wrote a lot on the love of the prophet Muhammad for his wife. As a meticulous historian, Mr. Deuel determined the authenticity of that little book". (Enbar 1966, p. 25)

Isaac Yahuda's unsurpassable skills and Muslim scholars' deep admiration for his expertise testify to his Eastern identity as someone deeply immersed in the Arabo-Islamic culture in the East and invite us to scrutinize his Arab Jewishness as far as knowledge of Arabic and Islam is concerned. His proficiency in Arabic testifies to his belonging in the East and his historic connection with it. This is further evidenced by his interest in the character of the Arab Jew, as in the example of al-Samaw'al Ibn 'Adiyya', the community leader and poet who lived in pre-Islamic Arabia. His association with al-Samaw'al, affirmed by the latter's eloquence in Arabic, manifest in the compilation of fine poetry. Just like al-Samaw'al, Yahuda demonstrated an interest in Arabic poetry, though not by means of production but rather by editing and disseminating it to the public, as in the case of publishing *Sharḥ al-Maḍnūn bi-hi 'ala ghayr ahlih* [That which is to be Withheld from Those Unworthy of It], an anthology of Arabic poetic verses accompanied by commentary by the fourteenth-century Muslim scholar 'Ubayd Allah Ibn 'Abd al-Kafī (al-'Ubaydī 1913–1915, p. introduction). Another manifestation of his Arab Jewishness was serving as a mediator between Jews in late Ottoman and British Mandate Palestine and the Arab world as declaring his connection to his birth town, Jerusalem, by calling himself *al-Maqdisi* (the Jerusalemite) in his book *Sharḥ al-Maḍnūn bi-hi 'ala ghayr ahlih* which he published with his private funds.[1]

### 3. Isaac Yahuda and Zionism

Isaac Yahuda actively integrated his "Arabness" into his Zionist engagement. His cultural, social, and economic attachments to Middle Eastern localities and his admiration for Arabic and Islamic culture exemplified his belief that Zionism could be enriched by the indigenous cultures of the region, rather than being exclusively a product of European Jewry and an embodiment of European ideals. Isaac Yahuda's entry into Zionism was made possible due to a few factors. First, the religionization of Zionism by several emissaries who imposed their understanding of Judaism through Zionism, particularly the messianic view of the ingathering of the exile, and found in Arab Jews a strong religious favor (Shenhav 2006, p. 78). Yahuda's religiosity was also a factor in his engagement in the Zionist enterprise. From an early age, he studied the Bible and the Talmud as well as rabbinic literature. Later in his life, he contributed commentaries on selected biblical passages, underscoring the benefits of the Arabic language to advance a better understanding of the Jewish scripture (Ben-Ze'ev 1941; Yahuda 1904). It is likely that Isaac viewed Zionism as a messianic movement that would redeem the Jewish people from the diaspora and bring them to their ancestral homeland (Itzhak 2007). Another factor behind his inclination towards Zionism is the desire for the unification of the fragmented communities dispersed throughout the Ottoman Empire. In this regard, Isaac Yahuda's stance on Zionism resembled other Ottoman Jews who construed their participation in it would not contradict their loyalty to the Ottoman homeland (Campos 2010). He perceived it as a cultural Hebraist movement that would elevate the literary and cultural status of Ottoman Jews who spoke a variety of languages (Ladino, Greek, Arabic, Turkish, French), which he believed led to their division, and would connect them with their brethren coming from Europe (Cohen and Stein 2014, pp. 215–21). Therefore, he played an active role in the revival of Hebrew participating in earlier institutions that encouraged the dissemination of Hebrew like the Hebrew language committee founded by Eliezer Ben-Yehuda, and used Hebrew as a literary medium by which he wrote articles and books and into which he translated Arabic works. His early involvement in the movement of Hebrew revival challenges the view that Hebrew culture was an exclusively an Ashkenazi phenomenon and underscores the contribution of Arab Jews to this project (Levy 2013). Among these earlier advocates for the dissemination of Hebrew was his late wife Vaida, the daughter of the of the Moroccan Rabbi Shim'on Ashriki. Yahuda praises her for being the first woman among Mizrahi Jews

to not only speak Hebrew, but also raise her children in the language of her "people" and to her memory Yahuda dedicated his work on the Western Wall (Yahuda 1929). Aside from his Hebrew literary activities, Isaac Yahuda participated from an early stage in the political activities of the Zionist movement. He supported Jewish immigration from oppressive countries in Eastern Europe and Russia as an act of solidarity that would not interfere with his loyalty to the Ottoman homeland and perceived the Jewish settlement in Palestine as a way to realize the full economic potential of the land, similar to the position of other Ottoman Jews (Cohen and Stein 2014, pp. 215–21).

Isaac Yahuda's cultural and social activities connected with Zionism during the mandate period made him subserve his knowledge of Islam for the benefit of the Zionist enterprise and to bolster Zionists' claims over sacred sites in Jerusalem during a time of national tension between Zionists and Palestinian nationalists. While aligning his "Jewishness" with the aims of Zionism, his "Arabness", which he expressed in his cultural, social, and economic attachment to localities in the Middle East; Baghdad, Cairo, Jerusalem, and Morocco, his admiration of Arabic and contribution to the dissemination of Arabo-Islamic knowledge in the Middle East and Europe, and his social and cultural connections to Arab intellectuals in the region affected his relations with other luminaries in the region but did not cause rupture with them. In the midst of national controversies, he demonstrated an understanding of Islamic theological beliefs and repackaged them to dissuade Arab audiences from responding to the calls of leaders of nationalist movements and to convey to the Hebrew readers the contradictions of their Muslim opponents in their beliefs. He was taking an active role in interreligious debate with Muslims. Through the exploration of this debate, Isaac Yahuda continued to show his belonging to the Arab world by establishing a common ground with his opponents based on their belief system.

*3.1. Hebrew Engagement with the Quran*

With their interest in striking roots in the East, Hebrew writers utilized Arabo-Islamic culture in grappling with issues pertinent to the construction of selfhood, the retrieval of what they considered their "ancestral" homeland, the advancement of Hebrew national culture, and the recuperating the place of Jews in the East (Hussein 2018). In their engagement with the Quran, Hebrew writers reiterated the German Jewish Orientalist paradigm of underscoring the Jewish influence on the origins of Islam by highlighting the ideas, beliefs, and figures that found resemblance in the Bible, though these writers were carrying out this undertaking within the Zionist context informed by the Zionist ideals. The Quran, in this process, constituted a sublime authority among Muslims and non-Muslims in the East. The sacred texts for Muslims served as textual evidence of the ancient origins of Jews in the East and a testimony of their interactions with Arabs in pre- and the post-era of Islam. Complete and partial translations and interpretations of the Quran were published to quench Hebrew readers' curiosity over a period spanning from late Ottoman and Mandatory Palestine to modern-day Israel.

In the preface to his translation, Rivlin elaborates several factors indicating the significance of the Quran and its value to Hebrew readers: first, the dissemination of the Quran and its fundamentality for a multitude of people around the world; second, the ability of the Quran to unify seventh-century Arab tribes after a long period of blood feuds; third, Islam's success in building a community and fueling it with courage and power to control large swaths of land; and fourth, the powerful nature of the Quran that attracted and has continued to draw the attention of people in Asia and Africa (Rivlin 1963, p. v).

Highlighting the Semitic origin shared by Judaism and Islam was crucial in the formative period of the Hebrew culture to foreground commonality between Jews and Muslim Arabs in late Ottoman and Mandatory Palestine. Hebrew intellectuals made a conscious attempt to underscore their shared Semitic origins in the pursuit of the Quran and the dissemination of this knowledge among the Hebrew reading audience. The objective of the use of "the Semite chronotope", as Yoni Furas has suggested, was to consolidate the Jewish narrative of an ancient right to Palestine and to "demarcate cultural and geograph-

ical territory" (Furas 2020, p. 34). In addition to invoking the Quran to highlight Jewish indigeneity in the East and the racial affiliation with Arabs, Rivlin sees in the Quran a work that would restore the Semitic spirit of the Jewish people claiming historical rights to Palestine. Underscoring the connection between Jews and the Quran, Rivlin writes, "The Jews find in the Quran a special value since it is one of the most wonderful productions of the Semite spirit. It is replete with prophetic pathos characteristic of Semites. Also, the rhythm is identifiable with our ancient productions" (Rivlin 1963, p. preface). Rendering the Quran in biblical Hebrew in Palestine's 1930s, Rivlin emphasizes the need "of the awakening of the Jewish people to return to the Orient: to its spirit and its life" (Rivlin 1963, p. preface). Rivlin's linkage between the Jewish return to the Orient and the contemplation of the Quran is indicative of not seeing the Quran as antithetical to Jews returning to Palestine, let alone as anti-Semitic. A thorough reading of the Quran in Hebrew indicates the rooted place of Jews in the Orient, from his perspective. The use of biblical Hebrew and the Hebrew word *parasha*, a term used to refer to portions in the Pentateuch, indicate, from the perspective of Hanan Harif, the domestication of the Quran within Hebrew culture (Harif 2016, p. 48). These translation decisions, most significantly, refer to the construction of the Eastern cultural sphere where two texts are integrated through translation into one another. In fact, in doing so, consciously or unconsciously, Rivlin embraces one of the Quran's self-images as a *reminder* (al-Dhikr) of the intertwined world of Jews and Muslims.

### 3.2. Arab Jewish Polemicist

Isaac Yahuda's deep grounding in religious studies and commitment to Judaism naturally inclined him towards embracing Zionism as the movement emerged in late Ottoman Palestine. As the Ottoman Empire waned and gave way to British mandate rule, Yahuda's religious devotion provided a conduit for him to integrate into Ashkenazi Zionist communities, which became more prominent during this period. Within the Mandate milieu, his religious background afforded him a notable voice within the ongoing Zionist discourse (Shenhav 2006, p. 77). His expertise of Arabic and Islam, similar to other Oriental and Sephardi Jews, was useful for mainstream Zionism and stressed Jewish claims to Palestine as a national home for Jews in light of the Balfour Declaration (Jacobson and Naor 2016). The disturbances of the Wall of 1929 in Jerusalem that harvested the lives of dozens among Jews and Arabs in Palestine signaled, according to some scholars, the beginning of the conflict between two communities given the violent nature of these confrontations that resulted from the conflation between religion and nationalism (Cohen 2013). These events, furthermore, constituted a formative moment in the construction of the Palestinian Arab national movement and had a considerable impact on the development of the Jewish Yishuv in Palestine and the establishment of many central ideas and beliefs that would shape the unique form of Zionist nationalism (Saposnik 2015). A central development that emerged from this decisive historical moment in the relations between Jews and Arabs was turning the Western Wall into a symbol of national identity, incorporating it into a growing national culture, and using it to create a new sense of sacredness that is associated with the nation (Saposnik 2015). Against the watershed of the transformation and the nationalization of such a religious icon, Isaac Yahuda utilized his knowledge of Islam to produce polemical discourse to underscore the indigeneity of Jews in the East, to protect the religious identity of his Hebrew readership, to undermine Muslims' claim to "contested" scared sites, and at the same time to demonstrate his authority among members of his community in the Yishuv and to show his loyalty to the Zionist enterprise. Rather than approaching his polemical commentaries on selected passages from the Quran as a form of verbal aggression against Islam and Muslims, his interaction with the Quran provides an opportunity to explore the political, social, and cultural dynamics in the Jewish-Arab relations in such a turbulent historical period in Palestine.

Isaac Yahuda was one of "the first Levantines", as Itzhak Bezalel noted, in late Ottoman Palestine (Bezalel 2011, p. 75). He was a descendant of a notable Baghdadi Jewish family that settled in Jerusalem in the early nineteenth century. As a Jerusalemite, the

intercommunal relations he experienced in urban communities in Jerusalem, Cairo, and Baghdad molded his character as an Oriental Jew. His character indeed encompassed elements from the intertwined Jewish, Arab, and Western worlds of the Eastern Mediterranean. These qualities are evident in Yizḥak Molcho's description: "His personality mixed the best qualities of Babylonians, Moroccans, Sephardim, Ashkenazim: the purity of heart and soul, honesty, modesty, industriousness, diligence and steadfastness" (Molcho 1943, pp. 18–19). Isaac Ben-Zvi, in his obituary of Isaac Yahuda, relates how the latter's personality reminded the Ashkenazi Jewish immigrant of the indigeneity of the Jewish people in the East (Ben-Zvi 1942). In his polemical approach, as an Oriental Jew, Isaac Yahuda's works position him within a longstanding tradition of Jewish-Islamic intellectual exchange. In this regard, Isaac Yahuda connects himself to medieval Jewish scholars, like Moses Maimonides and Abraham Maimonides, who, while being part of the Islamic world, engaged critically with Islamic thought, blending elements of agreement with polemical discourse (Boušek 2011; Russ-Fishbane 2013). Yahuda's perspectives demonstrate the continuity between his polemical style and those of renowned medieval Jewish scholars in the Islamic world.

In his reading of the diaries of Wasif Jawhariyyeh, an Arab Orthodox Jerusalemite musician and a contemporary of Isaac Yahuda, the Palestinian historian Salim Tamari suggests approaching Jerusalem and its various quarters as an embodiment of modernity and not the outcome of confessional division. The social structure was "fluid", and inhabitants of the various religious persuasions were not confessionally bound to their space but enjoyed liberal freedom moving between various quarters (Tamārī 2005, p. 28). One of the factors that caused the Arab population in pre-World War Palestine to identify with the country was a strong and local attachment to place, as Rashid Khalidi has indicated that the local population in Palestine had a powerful tradition of urban patriotism similar to what was prevalent in other Islamic cultures (Khalidi 1997, p. 153). Isaac Yahuda demonstrated his attachment to Arab cities, including Cairo, Baghdad, and Jerusalem, in signing his name at the bottom of the introduction to the work *sharh al-madhnun bihi 'ala ghayr ahlihi* published in Cairo in 1913. While he references his Iraqi origins, he takes pride in being "*al-maqdisi*" (the Jerusalemite) by birth, an act that resembled that of other residents in Palestine including those in Jerusalem, Nablus, Jaffa, and Gaza. This use in modern times was perceived as an identifying element of shared identity of those who lived in the city in that they shared the same fate with other residents and were different from those who were outsiders, perhaps Ashkenazi Jews.

While it is true that the confessional boundaries in the city "were more porous than impermeable", they were not absent from late Ottoman Jerusalem, where Isaac Yahuda lived. As Michelle Campos has shown, regional developments such as World War One, the emergence of ethnic nationalist movements, and European colonialism had devastating consequences in Jerusalem, where religious violence bore some religious components (Campos 2021, pp. 67–68). At this turbulent time, Yahuda remained invested in creating bridges between Jews and Arabs, demonstrating his Arab Jewish identity and utilizing his knowledge and connection with both communities.

The Nationalization of a Religious Icon

As opposed to Zionist fractions like revisionists and religious Zionists who vociferously called for taking over the Western Wall in the wake of the 1929 disturbances and the Marghribi neighborhood to expand the plaza before the Wall to accommodate Jewish visitors, mainstream Zionism viewed Jewish interests at that moment in maintaining peace and not with escalation. The preservation of peace, as historians of Jewish-Arab relations have shown, was only a means to allow Zionists to realize their strategy of "peaceful penetration" (Cohen 2015; Porath 1974). The maintaining of peaceful relations with Arabs without stirring their religious sentiments even as both communities were contesting the ownership of such a holy site as the Wall necessitated sustaining a calm tone in the deliberations. Isaac Yahuda emerged as a suitable candidate to carry out this task. In a lengthy article published in *meassef Zion* and later on as a separate monograph on *ha-kotel ha-ma'aravi* (the

Western Wall), Isaac Yahuda took it upon himself to disrupt the status quo in favor of Jews by consolidating Jewish claims to the Wall while utilizing his Islamic knowledge.

First, in his monograph, Yahuda demonstrates familiarity with and understanding of the sacredness of the site in accordance with Islamic beliefs and sources. Through making connections between Islamic references on *al-haram al-Sharif* and Jewish texts indicating the sacredness of *ha- ha-bait* (the Temple Mount), he implicitly points out that the origins of the sacrality of the Wall are to be found in Jewish history and Judaism in a way to underscore the Jewish influence on Islam. In doing so, he follows the Orientalist research of late nineteenth- and twentieth-century German Jewish scholars who argued for the influence of Jewish sources, ideas, and beliefs on the development of Islam. Isaac Yahuda was somehow knowledgeable of German due to his sojourn in Germany for two years before he had settled in Cairo in 1906 (Ben-Ze'ev 1941). He kept himself abreast of the publication of German Jewish scholars on the origins of Islam and he utilized his knowledge of Islamic sources to further advance their claims in the Zionist context. In the debate between both communities in the aftermath of the events, both sides harkened back to their respected sacred texts to support their stance on the Wall and to protect the religious identity of their respective communities. Interestingly, while Quranic verses were meant to support the claims of Palestinian Arab nationalists and their audience, Isaac Yahuda took it upon himself to participate in this discourse by providing a different viewpoint that bolstered Jewish arguments about the Wall. A central and often cited Quranic verse in this regard is [17:1], which reads as follows:

> Glory be to Him Who carried His servant by night from the Sacred Mosque to the Farthest Mosque, whose precincts We have blessed, that We might show him some of Our sign. Truly He is the Herer, the Seer. (Nasr et al. 2015)

Whereas his Muslim opponents cited this Quranic verse to indicate the sacrality of *al-Haram al-Sharif* (the Noble Sanctuary), which Jews referred to as the Temple Mount, (Cohen 2017) and to demonstrate its central place in their faith, history, and tolerant relations with other religious communities, Isaac Yahuda highlights that very sacrality as a testament to Jewish impact on Islam. Similar to Western Orientalists who viewed the similarities between the Quran and the Hebrew Bible as evidence of the former's dependence on and borrowing from the latter and a concerted effort by the early Muslim community to appropriate Jewish and Christian learning (Pregill 2008), Isaac Yahuda directs his energy to reveal this appropriation dwelling on the Quran, *hadith*, and *tafsir*. In doing so, he did not see himself degrading Islam considering claims made by the Islamic tradition itself that connect its sacred text and Muhammad to previous scriptures and prophets. The Quran is portrayed as a confirmation of previous divine revelations and continuation of the core message of the Bible [Quran 35:31; 46:30] and Muhammad is depicted as the long-awaited messiah that was mentioned in both the Old and New Testaments [Quran 7:157]. In his commentary on [17:1], he directs his multilinguistic skills and religious expertise to pinpoint the origins of the sacrality of *al-Haram al-Sharif* in biblical and extrabiblical sources.

In his interpretation of the cause for the sacrality of the *al-Haram al-Sharif*, Isaac Yahuda resorts to medieval Islamic tradition by invoking *Isra'iliyyat* (the Israelite lore) in discussing the causes connected with granting this place its special status. In commenting on the Quranic verse "whose precincts We have blessed", in referring to "the farthest mosque", Yahuda connects it to the song of ascents in psalms. The fifteen short psalms have been explained by scholars as referring to the centrality of Jerusalem, Zion, and the Temple (Berlin et al. 2004) and to Yahuda that suited the subject of underscoring the centrality of the site known to Muslims as *al-haram al-sharif* and to Jews as *har ha-bayit* (the Temple Mount). In particular, Isaac Yahuda connects the blessing mentioned in the Quranic verse to [Psalms 133:3], which reads "There the Lord ordained blessing, everlasting life". In comparing the Qur'anic reference "whose precincts We have blessed" with the psalmic partial verse "there the Lord ordained blessing", Yahuda illustrates the similarities between both scriptures in ascertaining the blessedness of the site as a divine imperative.

Isaac Yahuda's commentary was produced in Hebrew and meant to protect the religious identity of his respective community and bolster their religiously nationalized claims to the Wall. At the same time, his discourse did not negate the significance of the Wall for Muslims nor did it ridicule or undermine the accounts associated with the sacrality of *al-Haram al-Sharif*. In fact, he dedicated in his monograph a detailed discussion of *al-israa wa al-mi'raj* (the Night Journey) in Islamic tradition and even examined the different opinions among Muslims whether the event was a spiritual or a physical journey without arguing for one opinion over the other, though the dominant viewpoint among Muslims is that the night journey was a miraculous event that occurred to Muhammad physically and spiritually. He consistently refers to Muhammad in the account as either "the prophet" or "their prophet" without adopting disparaging references. That is understandable on his part since Muslims themselves respect Jewish prophets and relate to them with dignity and reverence. Isaac Yahuda's respectful view correlates with a common attitude among Yemenite Jews towards Muhammad as a prophet for the Arabs. The twelfth-century Yemenite Jewish philosopher and theologian, Rabbi Nathanel Ibn Fayyumi, who penned the work "*Bustan al-'Uqul*" (Garden of the Intellects), acknowledged the prophethood of Muhammad, affirming him to be a true prophet and a messenger of God. However, Ibn Fayyumi asserted that Muhammad's prophetic mission was specifically directed at the Arab people and was not meant to extend to the Jewish population (Ahroni 1998, pp. 55–56).

One of the questions Isaac Yahuda discusses in his monograph on the Wall is: from which gate did Muhammad enter the temple mount? Muslims claim that he entered the site from the Western gate and to the Western Wall either he or the archangel Gabriel tethered the winged horse to the Wall and hence the Wall acquired the famous name the *al-buraq*. Isaac Yahuda disregards this account and, in its stead, explains that earlier Muslims believed the location from which Muhammad entered the city was from the eastern gate. Why specifically the eastern gate? Because he thought that earlier Muslims and Jewish converts believed that Muhammad was the messiah and since the eastern gate to the Temple Mount would be the one from which the coming messiah would enter the city on the top of a white horse, Muhammad must have entered the city and the site from the eastern gate which is called as *bab al-rahma* (the mercy gate) or (the golden gate).

The knowledge and the dedication Isaac Yahuda put in his work *ha-kotel ha-ma'aravi* (the Western Wall) provide unprecedented service to Jewish claims to the Western Wall. Indeed, the Jewish Agency utilized this work and presented it to the British government in Palestine in 1930 to support its claims, according to Isaac Ben-Zvi (Ben-Zvi 1942).

*3.3. Surat al-Fātihah and Psalms*

Despite their political and social ramifications in affecting relations between Jews and Arabs in Mandatory Palestine, the events of 1929 did not completely sever connections between both communities. Economic and cultural relations inside and outside Palestine continued to thrive between members of both groups. Isaac Yahuda had worked as a bookseller based in Cairo since 1906 (Ben-Ze'ev 1941). Through his business activities, he forged a wide network of social, cultural, and economic relations with renowned booksellers and publishers in Egypt and other Middle Eastern countries. Muhammad Amin al-Khanji (1865–1939), a key player in the trade of books and manuscripts, founded his bookstore near al-Azhar in the commercial-residential area of al-Jamaliyyah in 1885. Over his career, he published hundreds of books (Zirikli 1997, vol. 6, p. 44). A pivotal figure, al-Khanji collaborated with Isaac and Abraham Yahuda, connecting them with a network of booksellers and publishers across the Middle East and North Africa (Gonzalez 2020).[2] Yahuda's connections extended beyond the world of books and manuscripts trading to forge cultural and intellectual ties with Nahḍa luminaries, including Ahmad Taymur Pasha, Ahmad Lutfi al-Sayyid, and Ahmad Zaki Pasha, who made efforts in the revival of Arabo-Islamic literature (Ben-Ze'ev 1941). Along with his brother Abraham Shalom Yahuda, both had continuous successful business dealings with a robust network of books and manuscripts dealers in the Middle East that supplied them with materials that Yahuda

sold to European institutions. The almost two-decade exchange of letters between Yahudas and al-Khanjis testifies to the multidimensional relations between both sides that went beyond bossiness dealings to include also social relations that had never been fully served due to the political circumstances in Mandate Palestine.[3]

In parallel, intellectual engagement between Arab Jews, like Isaac Yahuda, and Islamic thought, particularly the Quran, continued and was reflected in his writings. Isaac Yahuda's interest in the Quran was informed in part by the investigation of ethnographic and folkloric materials as looking at the ways in which Muslim communities in the East interacted with the scripture in their daily life. Unlike romantic Orientalism that aimed at the introduction of Arabs' perspective and their manners through the lens of alterity, Isaac Yahuda's rhetoric which echoed by other Sephardi and Oriental Jews including David Yellin, Abraham Shalom Yahuda, and Yosef Meyuhas, aspired, as Amos Noy has articulated, at revealing the need of a cultural and moral awakening based on the culture and the values of the East (Noy 2017). Isaac Yahuda in fact went beyond the ethnographic and folkloric materials associated directly with Jews (Evri 2020, p. 259), to study aspects of Arabo-Islamic culture from an ethnographic point of view. One way to assess Isaac Yahuda's cultural enterprise in the articulation of the linguistic and cultural connections between Arabo-Islamic and Jewish cultures is through his engagement with the Quran.

Isaac Yahuda included the findings of his ethnographic research in his seminal work "*Mishle 'Arav*" (Arabic Proverbs), a book in three volumes, the first of which was published in 1932 by the Palestinian Association for History and Ethnography. Unlike works whose authors concentrated on the collection of proverbs of certain ethnic, religious, or national communities, Isaac Yahuda's *Mishle 'Arav* was a comprehensive ethnographic work that comprised proverbs used among all communities of the East, regardless of their religious affiliation, ethnicity, or social class. The author, as he said himself, documented any maxim or popular saying he had chanced to hear or come upon in works throughout his voyages and the life he lived in the East. Consequently, *Mishle 'Arav* provides us with a wealth of examples attesting to the rich amalgam of cultures that formed a mosaic in the East of which Yahuda was part. Passages from the Quran are integrated in the work mainly to provide the cultural and religious context for proverbs or customs Isaac Yahuda relates to in his work.

Surat al-Fātiḥah (the Opening) appears in *Mishle 'Arav* in the context of explaining a Muslim custom of invoking the surah during a visit to the tomb of a righteous saint to seek his intercession (Yahuda 1932, p. 206). Ethnographic interest in local customs and traditions of people in Mandatory Palestine, including visiting tombs of saints, aimed mainly at the recovery of native customs of communities in the East. While Western Orientalists had demonstrated a scholarly interest in the subject in their quest to study "antiquities of the Holy Land", in their quest to justify colonialism and cultural dominion (Martin 1990), native Jerusalemite writers attempted to "modify" "Orientalist discourse in favor of finding a niche within its confines" to address issues pertaining to their nativity (Tamārī 2004b, p. 27). A major figure in this movement of Jerusalemite ethnographers was Tawfiq Canaan, who was preoccupied with "peasant religiosity and belief patterns" (Tamārī 2004b, p. 27). Isaac Yahuda was similarly attentive to Muslims' belief patterns with regard to their relationships to tombs of saints. In commenting on this popular custom, he provided a Hebrew translation of al-Fātiḥah, a commentary on it, and several critical points. Known in Arabic as al-Fātiḥah, this seven-verse chapter is considered by scholars of Islam as the most important chapter. The placement of the relatively short surah before the general progression of the chapters from the longest to the shortest suggests that it is meant to be recited before commencing reading the Quran (Reynolds and Qarai 2018, p. 30). It also functions as the introduction to the book, for it comprises the main ideas scattered throughout its various chapters, such as the oneness of God, the divine characteristics, the belief in the hereafter, and the categorization of mankind into different categories based on their belief in and denial of the divine message. Furthermore, this surah is recited in

every prayer a Muslim performs, on average seventeen times a day, and is also recited to strengthen one's faith and belief in the oneness of God[4].

## 4. Affinitas between Al-Fātiḥah and Psalms

In comparing the form of Quranic surahs and biblical writings, Angelika Neuwirth recommends attending to the similarities and dissimilarities between the scriptural texts. With respect to the latter, she stresses that in "the different phases of its genesis it [the Quran] comes very close to particular partial corpora". She explains that in its earlier stages, the Quran corresponds to psalms in its form (Neuwirth 2019, pp. 89–90). In his commentary on al-Fātiḥah, Isaac Yahuda also remarks on the connection between the surah and psalms with respect to the form.

Yahuda also finds similarities between al-Fātiḥah and psalms in their function. In his perspective, both are generally recited at "the beginning of everything" (Yahuda 1932, p. 206). Besides the ritual reciting of the surah at the beginning of prayer, it is also recited for healing and protection, according to some Islamic traditions. Similarly, according to (Shevuot 15b) in the Talmud, Jews have recited psalms for much of their history for protection. Yahuda finds in the invocation of the surah at the tomb of the saint a similar function and practice for psalms in the Jewish culture (Smith 1993; Yahuda 1932, p. 206). In this regard, he mimics other Jerusalemite ethnographers who saw in the practices of the native population of the East evidence for biblical affinities, a shared legacy (compare his works with Stephen in Tamārī 2004b, p. 31).

Besides form and function, al-Fātiḥah and psalms are similar in style. The psalmodic style of the surah is manifest in internal and communal dialogue. The former involves a conversation between God and the believer. According to hadith qudsi, a divine tradition that designates a direct discourse statement ascribed to God, the surah is a dialogue between the reciter and God. According to hadith qudsi, "I have divided the prayer into two halves between Me and My servant and My servant will receive what is requested. When the servant says: Praise belongs to God, Lord of all beings, God says: My servant has praised me. When the servant says: The Merciful, the Compassionate, God says: My servant has lauded Me" (Rippin and Knappert 1990, p. 76). With regard to communal participation, the surah is recited in prayer a few times a day and concludes with "Amen" (Āmīn), which Yahuda includes in the translation of al-Fātiḥah, similar to its use in the Jewish and Christian traditions (Akasoy 2019).

Isaac Yahuda's translation and commentary on al-Fātiḥah highlight the affinity between the Quran and the Hebrew Bible on the thematic level. The shared themes include singing the praise of God, seeking his help, pursuing his trust, looking for his guidance, and begging for his deliverance. Unlike the Book of Psalms, where there are no individual psalms that could touch on all the Book's themes (Neuwirth 2019, p. 241), al-Fātiḥah, as indicated by Yahuda is *'em ha-sefer* (the mother of the book) (Yahuda 1932, p. 206). Yahuda's description of the prominent Quranic chapter as "the Mother of the Book" acknowledges its essential place within the Quran. This term, which stems from Hadith traditions, signifies the chapter's position as the first in the Quran and its prescribed use as the starting point for Muslim prayer. In Arabic linguistic tradition, the phrase "mother of" denotes the source or essence of something. Consequently, the expression "the Mother of the Book" symbolizes the foundational nature of the chapter within the Quran. That is the reason Yahuda abstained from highlighting an individual psalm to articulate the connection between two scriptural texts; rather he indicated the psalms in general. Also, while individual psalms are chosen to address a specific issue from a liturgical point of view, al-Fātiḥah is used to function as such in various settings.

One of the themes dominant in the prevalent discourse among the communities of the Quran in the early Meccan period, as highlighted by Neuwirth, was consolation (Neuwirth 2019, p. 241), which is found in both al-Fātiḥah and psalms. "Thee we worship and from Thee we seek help" [1:5] and "guide us upon the straight path" [1:6] are similar in style

to individual psalms like, for example, [Psalms 3], where David asks God for help and [Psalms 4], where the psalmist highlights his confidence in God more than his troubles.

Writing about the relationship between the Quran and the Bible, Taha Hussein indicated their interrelatedness in the manner in which the Quran addresses itself as a complement to the Jewish and Christian traditions (Hussein 1958, p. 249). A manifestation of this interrelatedness is the emphasis on monotheism by the three Abrahamic religions; when they "speak of the monotheist God, that affirmation extends to all those who concur on the normative and indicative traits of "God in general" (Neusner 2012, p. 27). Islam and Judaism share the conviction that God is one as well as the belief in prophets, scriptures, beliefs in angels and devils, and communal accountability and responsibility (Esposito 2011, pp. 6–7). In addition to the general connection between al-Fātiḥah and psalms, Yahuda chooses to render the divine name "Allah" into Yhwh. What might be the psalter's theological center that rendered itself in his choosing Yhwh to translate Allah instead of Elohim? The combination of God as middat ha-din (the attribute of divine judgement) and middat ha-rahamim (the attribute of divine compassion and love) guides us. Middat ha-din (attribute of divine judgment) is stressed in interpreting the center of the Psalter as "The Lord reigns"; he is the judge and the sovereign. Yet, this sovereignty is exercised through love, which gives voice to middat ha-rehamim (the attribute of love) (See various interpretations of the theological center of the Psalter in: DeClaissé-Walford et al. 2014, pp. 44–45). The attainment of God's sovereignty through love is also expressed in al-Fātiḥah, where the name of God as compassionate and merciful is repeated twice in addition to invoking his divine quality as he who guides one to the straight path and as he who provides help for seekers. After all, God's mercy, in Islamic theology, precedes his wrath. The perception that God's compassion takes precedence over his force made Yahuda choose Yhwh as the translation for Allah in his Hebrew translation of al-Fātiḥah.

God is referred to in al-Fātiḥah as the source of compassion and as the master of the day of judgement. In fact, the surah begins with the exhortation of praise *al-hamd li-llah* (Praise be to God); yet, it has another side, representing God as the master of the day of judgement *malik yaom al-din* "the master of the day of judgement". The double manifestation of God as the source of praise and as that of power and sovereignty recurs in the Hebrew Bible in general and in psalms in particular. The double manifestation of divine working in the Hebrew Bible as highlighted by some scholars is manifested in the rabbinic language of God. It is a rule of the Rabbis that the tetragrammaton in the Hebrew Bible refers to *middat rahamim* as in [exodus 34:6] "YHWA 'el rahūm" and that *Elohim* refers to *middat ha-din* as in [exodus 22:8] (Kadushin 2001, p. 217). This combination is also found in psalms. In Psalms 11 and 12, *middat ha-rahamim*, i.e., God "is righteous; He loves righteous deeds; the upright shall behold His face"; in [Psalms, 11:7], he is the helper of the faithful [Psalms 12:2].

Another illustration of the similarities between psalms and al-Fātiḥah is the emphasis on the name of God. In al-Fātiḥah, God is referred to with *lafz al-Jalalah* (Word of Majesty), which is Allah, two times; the first in verse one, known as *al-basmala*, and in verse two. The combination of middat ha-rahamim and middat ha-din is expressed in the Hebrew translation of the surah. In the Shema, one finds a combination of the personal name of God Yhwh and the impersonal rabbinic name Elohim. The first observation one detects in reading the Hebrew translation of al-Fātiḥah is rendering God's name (Allah) in Arabic into the personal name of God (Yhwh). This is an assertion that both Jews and Muslims worship the same God, an assertion that stems from both defining God as "the omnipotent and universal God they worship as singular, unique, and manifestly one and not many beings or entities. That affirmation extends to all those who concur on the normative and the indicative traits of "God in general" (Neusner 2012, p. 27). In agreeing about God in abstract definition and "on some of the same scriptural and narrative traditions", Jews and Muslims establish "a sharing of convictions" (Neusner 2012, p. 27). The affirmation of a monotheist God is stressed in rendering Allah as Yhwh in the two places where the name

(Allah) occurs in al-Fātiḥah Isaac Yahuda. That translation decision also lays the foundation for a common ground between the Quran and the Hebrew Bible.

No less significant than the choice to render Allah as is the concurrence on some of the indicative traits of God. In Shema', a daily Jewish prayer, the personal name of God (YHWA) is associated with intimacy and mercy "love the God your Lord", which corresponds to the meanings associated with Allah in al-Fātiḥah as merciful and compassionate; a meaning that is stressed twice in a short surah. In the *basmala* at the very beginning of al-Fātiḥah, the text reads "in the name of God the compassionate, the merciful". Not so long after that, one reads again "the compassionate, the merciful", two divine names which are intensifications of "mercy" or "loving-mercy". It is interesting to see here that Yahuda renders the two divine names *al-Raḥmān* (the compassionate) *al-Raḥim* (the merciful) in Hebrew as *ha-raḥmān* and *ha-raḥom*. As scholars of the Hebrew Bible have shown that neither merciful nor compassionate are among the common names by which God is represented in the Hebrew Bible, the Hebrew rendering of these two divine names suggests that Yahuda approaches the Quran in some ways as a commentary on the Bible. In this case, the Hebrew rendition gives more obvious names of God's mercy and love through intensifications.

## 5. Exegetical Approaches

Yahuda's exegetical observations on al-Fātiḥah are an epistemological disruption of a traditional reading of the seventh verse of the surah. The hermeneutical reading assumes a meaning that lies outside the text and is unattainable without mastering the ins and outs of Arabic language. In other words, attaining the meaning of the Quranic text is contingent on what is within the text as well as external knowledge of the occasions of revelations (*asbāb al-nuzūl*), knowledge of prophetic traditions (hadith), and other traditional accounts. In his straddling of the language and the context in reading the seventh verse of al-Fātiḥah, Isaac Yahuda approaches the text as evidence for the presence of Jews and Christians in the Near East, if not in Arabia in particular, "along with their scripture and traditions" (Griffith 2013, p. 55).

In his commentary on the Quranic verse, Yahuda reconciles his beliefs and those accepted in Islam and highlighted in Islamic commentaries. Along with the previous verse the seventh verse reads as follows: "(6) Guide us upon the straight, (7) the path of those whom Thou hast blessed, not of those who incur wrath, nor who are astray" (Nasr et al. 2015, p. 5). Quranic commentaries yield several readings of these two verses. According to some, *those who incur wrath* refers to idolaters and *those who are astray* refers to hypocrites. Others relate the terms to two categories of sin in which disbelief incurs divine wrath while lesser sins result from being astray. According to others, the two terms can be understood "as a reference to all various degrees of those who are spiritually debased and lost and preoccupied with procuring some share of the fortunes of this world" (Nasr et al. 2015, p. 5). As the final verse of the chapter, it can be understood as a prayer to follow the way of those whom God has blessed and to avoid the way of the disbelievers, some of whom are astray and with some of whom God is wroth. However, there is another controversial reading in which the terms *those who incur wrath* refers to Jews while the term *those who are astray* refers to Christians. It is this reading that triggered the attention of Isaac Yahuda, who abstained from referencing various readings among Muslim commentators, perhaps preferring to highlight the popular understanding of that Quranic verse or the possibility of adopting a polemic tone in relating to Hebrew readers about their negative perception in the Quran.

Yahuda's literary skills, accompanied by his Jewish learning, reveal themselves when grappling with the association between Jews and the divine wrath. The number of Muslim commentators who attribute the two terms to Jews and Christians rely on a prophetic saying, whose soundness is a subject for discussion. Nevertheless, Yahuda seems to accept the soundness of this prophetic saying without question given the common acceptance among Muslims. In his assessment, however, this association between Jews and divine wrath does not originate in Islam but is rather an embodiment of the Judeo-Christian polemical

relationship, which penetrated the corpus of Quranic commentaries through Arabs' contact with Jews and Christians in the eastern Mediterranean. In other words, it is an embodiment of the multifaceted relations between the people of the East that accommodated religious differences and accepted them. This polemic relationship characterized the Eastern world and survived in the Quranic commentaries[5]. In a way, Yahuda's explanation absolves Quranic commentaries from imposing the connection between God's wrath and Jews and highlights it as a reflection of the theological tension between Jews and Christians in the East. He even illustrates this further by explaining that the association between Christians and those "who are astray" according to the surah reflects the Jewish depiction of Christians found in Jewish sources (Yahuda 1932, pp. 207–8).

In his commentary on the Quranic verse, Isaac Yahuda not only creates an affinity with the language of the Hebrew Bible but also with biblical ways of thought. Although he does not explicitly read the Quranic verse in messianic terminology, his reading contains religious elements that lend themselves to some theological reinterpretation. While examining the affinity between Jews and divine wrath, he does not criticize those Quranic commentators who highlighted the connection between *those who incur God's wrath* and Jews. To him, this linkage is the product of the multifaceted encounter between Jews, Christians, and Muslims living in the East, which encompassed familiarity with and acceptance of polemical dimensions in that rich relationship. Most importantly, this linkage had already been established in the Hebrew Bible, where the trope of God's wrath is repeated time and again. It is noteworthy that in Yahuda's highlighting of the connection between Jews and divine wrath, messianic allusions worked their way to the surface. In unpacking this connection, he explains that that divine wrath is evidenced in the expulsion of Jews from their ancestral homeland, which is Palestine (Yahuda 1932, p. 206). How this connection is linguistically phrased in the Quranic text, he notices, indicates the continuous status of divine wrath, which indicates a messianic inclination that has been translated into his earlier support of Hovevey Zion—a forerunner of the Zionist movement that included religious Jews among its activists—regarding the Jewish settlement in Palestine and the dissemination of Hebrew (Itzhak 2007, p. 147). His comments on the association between Jews and God's wrath resonated with his beliefs and gave expression to his perception of Zionism and attachment to Palestine. Isaac Yahuda viewed God's wrath as a necessary step on the journey to divine deliverance and final redemption.

## 6. Conclusions

Isaac Yahuda's work offers an opportunity to reveal the rich cultural atmosphere of the Arab and Jewish world that once existed. His works are illustrative of the linguistic and cultural multiplicity that gave expression to the Eastern culture, a culture that existed from ancient times and survived to modern times until its suppression through the essentialization and ethnicization of the Arab and the Jew that dominated the hegemonic discourse in the Middle East due to the emergence of new realities. Revising this history of cultural interaction between Jews and Arabs through the lens of native Jews provides a source for cultural possibility in the future between the nations of the modern Middle East. Analyzing earlier Jewish engagement with Islamic scripture has shown the linguistic and cultural multiplicity of the Eastern culture and the acceptance of and familiarity with polemical dimensions in the Eastern world. Exploring translational and interpretational methodologies adopted by Isaac Yahuda demonstrates the extent to which the Quran was not read as an antithesis to the Hebrew Bible, but rather as complementing the biblical message.

Well versed in Arabic and Islamic culture, in addition to paying attention to the specificities of Islam and other monotheistic religions, Isaac Yahuda abstained from using language loaded with accusations towards Islam and its portrayal of Jews and Christians. Rather, he provided a perspective whose framework suggests that Islam's distinction between itself and other religions was couched in a terminology employed by previous religions, and thanks to the contacts between Islam and these religious spheres this termi-

nology evolved and took a different shape to address internal and external issues related to the relationship between Islam and the other Abrahamic religions.

Isaac Yahuda's monograph on the Western Wall aimed to consolidate Jewish claims to the site by utilizing his Islamic knowledge. Yahuda demonstrated familiarity with and understanding of the sacredness of the site in accordance with Islamic beliefs and sources. He made connections between Islamic references to *al-Haram al-Sharif* (the Noble Sanctuary) and Jewish texts indicating the sacredness of *Har ha-Bait* (the Temple Mount), implicitly pointing out that the origins of the sacrality of the Wall are to be found in Jewish history and Judaism. He utilized his knowledge of Islamic sources to advance further the claims of German Jewish scholars on the origins of Islam in the Zionist context. Yahuda directs his energy to reveal Islamic appropriation of Jewish learning, using the Quran, hadith, and tafsir. While doing so, he did not see himself as degrading Islam considering claims made by the Islamic tradition itself that connect between its sacred text and previous scriptures and prophets. Yahuda's commentary aimed to protect the religious identity of Jews and to bolster Jewish arguments about the Wall, highlighting similarities between Islamic and Jewish scriptures to underscore the centrality of the site known to Muslims as *al-haram al-sharif* and to Jews as *har ha-bayit*.

In commenting on al-Fātiḥah, Isaac Yahuda associates the surah with psalms. His association is brief, yet it invites us to contemplate the connection between these scriptural texts. The connection between the surah and psalms revolves around several discourses that reflect the internal debate between the monotheistic groups who lived in the Arabian Peninsula at the time of the Quran. The linkage is attested by the cross-references between the surah and psalms, which are attested in the use of a few topoi: the interpretation of the creation as the sign of God, the assurances of providence, and the judgement of the hereafter.

**Funding:** This research received no external funding.

**Institutional Review Board Statement:** Not applicable.

**Informed Consent Statement:** Not applicable.

**Data Availability Statement:** No new data were created or analyzed in this study. Data sharing is not applicable to this article.

**Conflicts of Interest:** The author declares no conflict of interest.

## Notes

[1]  It should be noted that the author of the roman, *Lidyah*, from which this story is extracted, is Isaac Yahuda's daughter Miriam Yahuda Enbar. In an interview, she revealed that her father's qualities influenced the literary figure of Yeshurun Deuel in her novel. For more details, see (Miriam Yahuda Enbar 1966).

[2]  Although the preserved exchange of letters between Muhammad Amin al-Khanji and Abraham Shalom Yahuda is dated to have started since 1926, references in the letters indicate earlier relations between both sides. Furthermore, al-Khanji likely introduced Isaac Yahuda to Muhammad Isma'il the owner of *al-Sa'adah* press in Cairo that published Yahuda's *sharh al-madhnun* in 1913–1915. Muhammad Amin al-Khanji had published his addendum to *mu'jam al-bildan* titled *kitab manjam al-'imran fi al-mustadrak 'ala mu'jam al-bildan* in 1907 with *al-Sa'adah* press.

[3]  The exchange of letters part of it is archived at the national library of Israel and part of it preserved with Muhammad Amin al-Khanji, the grandson of Amin al-Khanji to whom I am grateful for sharing these documents with me for research purposes.

[4]  In a work looking at the shared commonalities between the Hebrew Bible and the Quran, the Israeli writer Almog Behar began the project "Quran Yehudi" (Jewish Quran), in which he looks at agreement verses between both scriptures. In his rendering of the first chapter, al-Fātiḥah, interestingly, he abstains from translating the seventh verse as if it is the one that separates Jews from Muslims. For more details, see https://www.haaretz.co.il/literature/.premium-1.4304232 (accessed on 9 December 2019).

[5]  This tension is by the way sensed in *Jacob's Ladder* where Alice Gaon discovers that Donia the Lebanese Christian maid converted to Islam. See (Kamal 2017).

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
