# Peer review of "An Arab Jew Reads the Quran: On Isaac Yahuda’s Hebrew Commentary on the Islamic Scripture"

_religions, doi:10.3390/rel15040495_

Round 1

Reviewer 1 Report

Comments and Suggestions for Authors

Dear author,

Thank you very much for his interesting work. I think he has many good ideas and that the topic is an original contribution to scholarship. I think perhaps one of the elements that could be improved is the presentation of Yahuda's life. Now, it is talked about on page 3 and also on pages 6-7. In particular, there is a paragraph between the l. 29-30 (pp. 6-7) describing him as one of the first Levantines. You should consider incorporating this bit more at the beginning, after all it doesn't add much to the section he is now in (An Arab Jewish Polemicist). The same occurs with the data on the work of Yahuda (l. 160) that could go on l. 100. Next, although there is a list of references, no specific references are given in the text itself and it is not clear to me whether this is intended to be this way from the beginning. In any case, I miss them in several places (l. 89-90; 240; 310; 480; 540; 630; 640). Some sentences are too long (l. 430 "The most prominent figure ... Africa"; l. 620 "However, there is ... Quran"). There are phrases that should be reformulated or ideas that need to be clarified/further developed: l. 260 "In the mandate... Ottoman state"; l. 320 "where religious violence bore some religious components"; l. 520 "unlike the Psalter, ... mother of the book"; l. 630 "...or possibly to adopt...Quran"; l. 660 "...which indicates a messianic inclination...Hebrew." There seem to be some typos (l. 170 "Zionism a messianic"; l. 200 "East; Baghdad"; l. 210 "author he continued"; l. 280 "scared sites"; l. 660 "inYahuda"; l. 690 "Wall that aimed"; note 16 (two times 2011); note 20 cursive missing; notes 41 and 56 capitalization in transliterations.

Best wishes

Author Response

Dear Reviewer, 

I would like to thank you for taking the time reading and commenting on the article. Your comments and suggestions have guided me to address several issues that I have not paid attention to. Here are a detailed response to the raised points in the review: 

I think perhaps one of the elements that could be improved is the presentation of Yahuda's life. Now, it is talked about on page 3 and also on pages 6-7. In particular, there is a paragraph between the l. 29-30 (pp. 6-7) describing him as one of the first Levantines. You should consider incorporating this bit more at the beginning, after all it doesn't add much to the section he is now in (An Arab Jewish Polemicist). The same occurs with the data on the work of Yahuda (l. 160) that could go on l. 100.

I agree that Yahuda's biography is important to the article but I opt against compiling it into one section at the very beginning of the article as suggested. Instead, I choose to weave biographical details into the narrative where they are most relevant to the discussion at hand.

The inclusion of some biographical elements in the section "An Arab Jewish Polemicist" aims to highlight the influence of Yahuda's Levantine heritage on his polemical style. By labeling him as "An Arab Jewish Polemicist," I seek to convey how Yahuda's mode of argumentation was symptomatic of the intellectual milieu of the Near East. Yahuda’s polemics involved articulating dissent against certain interpretations of Qur'anic verses while simultaneously maintaining a posture of respect for Islamic culture. By doing so, I, moreover, aim to connect Yahuda to a historical tradition of Jewish scholars, like Moses Maimonides and Abraham Maimonides, who, while being part of the Islamic world, engaged critically with Islamic thought, blending elements of agreement with polemical discourse. This comparison serves to position Yahuda's work within a longstanding tradition of Jewish-Islamic intellectual exchange. Admittedly, I have not articulated this comparison in the submitted version and I will address it in the revised version of the article. 

Next, although there is a list of references, no specific references are given in the text itself and it is not clear to me whether this is intended to be this way from the beginning. In any case, I miss them in several places (l. 89-90; 240; 310; 480; 540; 630; 640).

References were indeed missing in the manuscript and they will be added in the revised version.

Some sentences are too long (l. 430 "The most prominent figure ... Africa"; l. 620 "However, there is ... Quran"). There are phrases that should be reformulated or ideas that need to be clarified/further developed: l. 260 "In the mandate... Ottoman state"; l. 320 "where religious violence bore some religious components"; l. 520 "unlike the Psalter, ... mother of the book"; l. 630 "...or possibly to adopt...Quran"; l. 660 "...which indicates a messianic inclination...Hebrew."

These writing style issues as well as typo issues will be taken care of in the revised version. 

Reviewer 2 Report

Comments and Suggestions for Authors

Author Response

Dear Reviewer, 

I would like to thank you for taking the time reading and commenting on the paper. I am grateful for your comments and recommendation. Below are my responses to the points you suggested in your review: 

  1. Providing a little bit more background on the politics and religious dynamics in that time to emphasize the challenges that Isaac Yahuda faced and his significance.

That is highlighted in the section “The Nationalization of a Religious Icon” and the manner in which Isaac Yahuda challenged militant and religious Zionism by using his knowledge of dialogue to support peaceful penetration of Jewish settlers into Palestine through immigration and land purchase without escalating unnecessary tension.

  1. A clearer explanation on how Yahuda's commentary challenged Zionist ideals.

In the section titled "Isaac Yahuda and Zionism," I explore Isaac Yahuda's relationship with the Zionist movement and examine how he contested the notion of Zionism being solely a European construct and a manifestation of European ideals. I discuss how Yahuda's Arabness influenced his vision of a Zionism deeply rooted in indigenous cultural and social principles, which encompassed a respect for and integration of Arabo-Islamic culture. Also, on page 8 I show how his conception of Zionism challenged militant Zionism that during debates mobilized its supporters to occupy sacred sites in Jerusalem by force. This will be articulated in the revised version. 

  1. More examples of Yahuda's commentary, other than the two mentioned of Surat al-Fatiha and Surat al-Israa.

On page three on the manuscript, I referenced sources where the reader could find more examples of Yahuda’s interaction with other Qur'anic verses. I should indicate that he did not provide full commentary on other Qur'anic chapters other than al-Fatihah and verses from chapter 17 surat al-Israa.

  1. Please check that there are no missing references, for example: in line 500, there is no reference to the quote of Angelika Neuwirth.

Thanks for this. Upon revising the manuscript, I realized that many references were missing from the article. I added them throughout the manuscript.

Best wishes,